# Impact of Blood Loss on Renal Function and Interaction with Ischemia Duration after Nephron-Sparing Surgery

Stephan Buse [1,2] , René Mager [2,*], Elio Mazzone [3], Alexandre Mottrie [4,5], Sebastian Frees [2,†] and Axel Haferkamp [2,†]

1 Department of Urology, Alfried Krupp Krankenhaus, 45131 Essen, Germany
2 Department of Urology and Pediatric Urology, University Medical Center, Johannes-Gutenberg-University, Langenbeckstr., 155131 Mainz, Germany
3 Unit of Urology, Division of Oncology, URI, IRCCS Ospedale San Raffaele, 20132 Milan, Italy
4 Department of Urology, OLV Hospital, 9300 Aalst, Belgium
5 ORSI Academy, 9090 Melle, Belgium
* Correspondence: rene.mager@unimedizin-mainz.de
† These authors contributed equally to this work.

**Abstract:** Objectives: Nephron-sparing surgery (NSS) exposes the kidney to ischemia–reperfusion injury. Blood loss and hypotension are also associated with kidney injury. We aimed to test the hypothesis that, during NSS, both ischemia duration and blood loss significantly affect postoperative renal function and that their effects interact. Methods: Consecutive patients undergoing NSS were enrolled. The primary endpoint was renal function expressed as the absolute delta between preoperative and postoperative peak creatinine. We developed a generalized linear model with the ischemia duration and absolute hemoglobin difference as independent variables, their interaction term, and the RENAL score. The model was than expanded to include a history of hypertension (as a proxy for hypotension susceptibility) and related interaction terms. Further, we described the perioperative and mid-term oncological outcomes. Results: A total of 478 patients underwent NSS, and 209 (43.7%) required ischemia for a mean of 10.9 min (SD 8). Both the ischemia duration (partial eta 0.842, $p = 0.006$) and hemoglobin difference (partial eta 0.933, $p = 0.029$) significantly affected postoperative renal function, albeit without evidence of a significant interaction ($p = 0.525$). The RENAL score also significantly influenced postoperative renal function ($p = 0.023$). After the addition of a previous history of hypertension, the effects persisted, with a significant interaction between blood loss and a history of hypertension ($p = 0.02$). Conclusions: Ischemia duration and blood loss had a similar impact on postoperative renal function, albeit without potentiating each other. While the surgical technique and ischemia minimization remain crucial to postoperative kidney function, increased awareness of conscious hemodynamic management appears warranted.

**Keywords:** kidney neoplasm; nephrectomy methods; postoperative complications

## 1. Introduction

When feasible from a technical point of view, guidelines [1,2] recommend partial nephrectomy for the treatment of renal cell masses, as nephron-sparing surgery (NSS) appears to maintain renal function without impairing oncological results, not only for T1 tumors [3]. Accordingly, the use of NSS has rapidly increased during the last two decades, particularly in younger patients [4]. To improve access to larger, deeper masses, surgeons may have to clamp the renal artery during NSS, resulting in ischemic damage to the surrounding renal parenchyma. Three main pathways have been determined as triggers of ischemic damage [5]: endothelial damage resulting in vasoconstriction and vascular congestion, tubuli obstruction resulting from an electrolyte imbalance in the ultrafiltrate with cast formation and from cell debris, and reperfusion injury after declamping (e.g., mediated by reactive oxygen species and the associated inflammatory reaction). A systematic review addressing the impact of ischemia on renal function after NSS concluded that, in addition

to preoperative renal function and the volume of preserved renal parenchyma, an ischemia duration exceeding >20–25 min plays a crucial role in terms of postoperative and long-term renal function, not only in patients with a solitary but also in those with a bilateral kidney [6]. Notably, the presence of acute kidney injury after NSS has been associated with a reduction in long-term functional outcomes [7]. The impact of postoperative renal functional impairment was confirmed in a recent study comparing multiple models for the prediction of renal function impairment, estimated by a glomerular filtration rate decline at 1 year after partial nephrectomy: postoperative percentage glomerular filtration rate loss was a component among the 44 best predictive models, along with the ischemia technique and preoperative glomerular filtration rate [8].

Postoperative acute kidney injury is not only observed in renal surgery but is also frequently seen after major non-renal procedures. In this context, blood loss, hypovolemia, and the resulting hypotension have been recognized as the main triggers for perioperative renal injury [9–12].

The extent to which hypovolemia-triggered renal injury interacts with ischemia–reperfusion injury during NSS and affects postoperative renal function is underexplored. In this analysis, we address the hypothesis that, in patients undergoing NSS, both the ischemia duration and blood loss significantly affect postoperative renal function and that their effects interact.

## 2. Methods

This is a secondary analysis of data that were prospectively collected within a dedicated institutional database managed by the Urology Department of the University Mainz.

### 2.1. Study Population

Consecutive adult patients undergoing NSS for renal cell carcinoma (RCC) between 2009 and 2017 at the Urology Department of the University Mainz were enrolled in the institutional database that collected clinical routine data. According to Paragraphs 36 and 37 of the Hospital Law of the German region Rheinland-Pfalz ("Landeskrankenhausgesetz", § 36 and § 37), ethical approval and informed consent is not required for retrospective analyses of routinely collected data as long as anonymity is preserved.

### 2.2. Endpoints, Main Exposures, and Assessment of Variables

The primary endpoint was postoperative renal function, defined as the difference between preoperative and peak postoperative serum creatinine [13]. Postoperative creatinine was used as the highest available value (peak) before discharge. Further, we reported perioperative benchmarks, complications, and mid-term oncological outcomes.

The main exposures were ischemia duration, as extracted from the intraoperative clinical report, and blood loss, expressed as the absolute difference between preoperative and postoperative hemoglobin. Postoperative hemoglobin refers to the sample of the first postoperative day. All laboratory values were manually extracted from electrical clinical files and stored in a dedicated database. Trained personnel collected baseline demographics, intraoperative data, and postoperative complications from the clinical documentation of the index hospitalization and categorized it where applicable based on standardized definitions. Follow-up information was obtained during the routine oncological follow-up assessment, every 6 months during the first year and yearly thereafter [14].

### 2.3. Statistical Analysis

The analyses presented here followed a predefined analysis plan specifying the aims, population, primary and secondary endpoints, and statistical approach (including all confounders of the model). The analysis plan is available upon request.

We conducted a complete case analysis. The level of significance was at 2-tailed $p \leq 0.05$. We used SPSS 25 as statistical software.

Descriptive data are reported as the count (percentage with 95% confidence interval) and mean (standard deviation (SD)) or median (interquartile range (IQR)). Non-parametric data were described using the chi-squared test for binary and Mann–Whitney test for continuous data. We assessed the impact of ischemia, blood loss, and their potential interaction on postoperative renal function using a generalized linear model. Ischemia duration, absolute hemoglobin difference, their interaction term, and RENAL score were the independent variables. Further, to explore the assumption that blood loss was indeed an appropriate approximation of hypovolemia and hypotension, we added a previous history of hypertension and its interactions into the model, because hypertensive patients might be more sensitive to the negative effects of hypovolemia and hypotension resulting from blood loss [11]. Because of the nature of this analysis (secondary analysis of prospectively collected data), we did not conduct any formal sample size calculation.

## 3. Results

### 3.1. Baseline Characteristics and Index Hospitalization

This analysis was based on 478 consecutive patients, of whom 146 (30.5%) were women. The mean age was 63 (SD 12) years, and 196 (41%) patients were classified as ASA $\geq$ III. The preoperative median estimated glomerular filtration rate was 76.8 mL/min (IQR 59.2–91.5). A history of arterial hypertension was recorded in 224 (46.9%) patients (Table 1). Baseline characteristics did not differ ($p > 0.05$) in patients requiring and not requiring ischemia.

**Table 1.** Baseline and tumor characteristics.

| Baseline Characteristics | |
| --- | --- |
| Age, years | 63.5 $\pm$ 12 |
| Female sex | 146 (30.5%) |
| ASA $\geq$ 3 | 196 (41%) |
| Preoperative creatinine, mg/dL | 0. 94 (IQR 0.82–1.16) |
| Preoperative eGFR, mL/min | 76.8 (IQR 59.2–91.5) |
| Preoperative Hb, g/L | 14.2 (IQR 13.3–15.3) |
| History of hypertension | 224 (46.9%) |
| Histology and tumor size | |
| Clear cell RCC | 341 (71.3%) |
| Papillary RCC | 103 (21.5%) |
| Chromophobic RCC | 34 (7.1%) |
| Tumor size, cm | 3.5 $\pm$1.9 |

N = 1 missing histology.

An open technique was conducted in 321 (67.1%) patients. In 10 (2.1%) patients, the surgical technique was conventional laparoscopy; in 147 (30.8%), a robotic-assisted technique was used. The surgeon was a resident in 84 (17.6%) of the procedures, and surgery lasted a median of 180 min (IQR 150–221).

Only 27 (5.6%) patients presented a pT3 stage, while the vast majority were classified as pT1 ($n = 302$, 63.2%). The nodal stage was positive in 18 patients (3.7%), negative in 155 (32.4%), and pNx in the remaining 305 patients (63.8%). Metastatic disease was reported in eight patients (1.7%). The histology was clear cell RCC in 341 (71.3%), papillary RCC in 103 (21.5%), and chromophobic RCC in the remaining 34 (7.1%) patients (one missing). Mean tumor size was 3.5 cm (SD 1.9 cm), and RENAL score distribution was $\leq$6 in 39.6%, 7–9 in 43.6%, and >10 in 16.8%.

Surgical margins were positive in 22 patients (4.6%). Complications of Clavien-Dindo Class $\geq$II were registered in 24 (5%) patients, including 11 surgical revisions (2%), with seven (1.5%) due to postoperative bleeding.

### 3.2. Impact of Ischemia Duration, Blood Loss, and Their Interaction

Ischemia was necessary in 209 (43.7%) patients, with a mean duration of 10.9 min (SD 8; $\geq$25 min in 12 (5.7%) patients).

Median preoperative creatinine was 0.94 mg/dL (IQR 0.82–1.16; median 83 µmol/L, IQR 72–102), and median postoperative creatinine was 1.04 mg/dL (IQR 0.85–1.34; median 92 µmol/L, IQR 75–118). The postoperative in-hospital estimated glomerular filtration rate was 68.5 min/mL (IQR 49.8–86.6). Hemoglobin fell from a preoperative median of 12.6 g/L (IQR 13.2–15.3) to a median of 11.3 g/L (IQR 10.2–12.6) postoperatively.

Both the ischemia duration (partial eta 0.842, $p$ = 0.006) and hemoglobin difference (partial eta 0.933, $p$ = 0.029) significantly and strongly affected postoperative renal function, albeit without any indication for interaction ($p$ = 0.5). The RENAL score was also significantly associated with postoperative renal function (partial eta 0.896, $p$ = 0.023) (Table 2). After the addition of a previous history of hypertension and an interaction term between hypertension and blood loss (Table 2b), the impact of both the ischemia duration (partial eta 0.971, $p$ = 0.005) and blood loss (partial eta 0.992, $p$ = 0.008) persisted, without a significant interaction between ischemia duration and blood loss ($p$ = 0.126). A history of hypertension did not affect the difference between pre- and postoperative serum creatinine ($p$ = 0.870). In contrast, we detected a significant interaction between blood loss and a history of hypertension ($p$ = 0.026) with regard to postoperative renal function.

**Table 2.** Generalized linear model with delta creatinine (pre to post) as dependent variable; a. baseline model; b. model assessing the interaction between history of hypertension and blood loss.

| a. | Partial Eta-Squared | *p*-Value |
|---|---|---|
| Corrected model | 0.952 | 0.057 |
| Constant term | 0.749 | 0.000 |
| Ischemia | 0.971 | 0.005 |
| Delta Hb | 0.992 | 0.008 |
| RENAL score | 0.896 | 0.023 |
| Ischemia × Delta Hb (interaction term) | 0.692 | 0.126 |
| R-squared = 0.952 (corrected R-squared = 0.532) | | |
| b. | Partial Eta-Squared | *p*-Value |
| Corrected model | 0.995 | 0.013 |
| Constant term | 0.956 | 0.000 |
| Ischemia | 0.971 | 0.005 |
| Delta Hb | 0.992 | 0.008 |
| History (Hx) of hypertension | 0.006 | 0.870 |
| Ischemia × Delta Hb | 0.840 | 0.126 |
| Ischemia × Hx of hypertension | 0.001 | 0.951 |
| Delta Hb × Hx of hypertension | 0.852 | 0.026 |

R-squared = 0.995 (corrected R-squared = 0.873).

### 3.3. Oncological Outcome

After a mean follow-up of 15 months, 29 (6.1%) patients developed distant metastasis and 20 (4.2%) died. Of the 20 deaths, 7 (1.5%) resulted from cancer, and 13 (2.7%) were not cancer-related. Patients dying of cancer had, in 71% (5/7), multilocular metastatic disease, whereas metastasis was not present in any of the 13 patients who died from other causes. A robot-assisted technique tended to lower metastatic progression (2.8% vs. 7.6%), without reaching statistical significance ($p$ = 0.056), independent of surgeons' experience (Breslow-Day test: $p$ = 0.743).

## 4. Discussion

This study indicated that, firstly, postoperative renal function was significantly affected by a short ischemia duration (mean < 11 min); secondly, blood loss significantly affected postoperative renal function and its effect was in the same order of magnitude as the effect of the ischemia duration; third, the negative effects of the ischemia duration and blood loss did not interact, i.e., they did not potentiate each other. Finally, the significant interaction between blood loss and a history of hypertension supports the hypothesis that the impact of blood loss is mediated by hypovolemia and hypotension.

Volpe and co-workers [6] summarized the evidence on the impact of renal ischemia on postoperative renal function. While the data for a solitary kidney based on large samples (from 200 to 660 patients) repeatedly demonstrated an impact of ischemia duration on postoperative renal function, data were less consistent for bilateral kidneys, potentially due to the smaller sample sizes (approximately 60 patients, with a few exceptions) [6]. Of note, in the largest study, even a protracted warm ischemia duration did not negatively impact renal function [15]. In contrast, the evidence for the impact of preoperative renal function was consistent for both a solitary and bilateral kidney [6]. Our analytical approach using the delta between preoperative and peak postoperative creatinine concentrations inherently accounted for preoperative renal function. In contrast to previous evidence suggesting a cut-off of 25 min for ischemia to negatively affect renal function [6], we detected a significant impact of a shorter ischemia duration on postoperative renal function. On one hand, this might be the consequence of a larger sample size. On the other hand, the use of the difference between pre- and postoperative concentrations might have partially accounted for other factors and, therefore, allowed for the detection of the impact of even a short ischemia duration. Of note, postoperative renal function impairment was recently demonstrated as a major factor for a decline in glomerular filtration rate at 1 year after partial nephrectomy [8]. However, as suggested in a study comparing robot-assisted partial nephrectomy and percutaneous tumor ablation with a follow-up of 3 years or more, impaired 1-year renal function may show some recovery over time [16].

Our findings in terms of the effect of blood loss and hypovolemia/hypotension on renal function are in line with a large body of evidence on non-renal procedures [9–12]; in the renal surgery setting, however, this question, as well as the potential interaction between ischemia–reperfusion injury and hemodynamic disturbances as triggers for postoperative renal injury, does not appear to be explored.

The ischemia duration was low. Perioperative complications, positive resection margins, and survival were in the same order as in previous cohorts [17–19].

Strengths of our approach are the size of the study populations (*n* = 478) and a cohort enrolling consecutive patients based on broad eligibility criteria, thus reducing the risk of selection bias. We are aware of the following limitations. First, the data were generated at a single institution; however, relevant quality criteria, such as the complication rate, surgical margins, and renal function impairment, as well as oncological outcomes, were within the range of previously published data, supporting the generalizability of our results. Second, we estimated renal function using serum creatinine. Creatinine has been criticized as insensitive in terms of renal function and alternatives have been proposed. However, creatinine remains the most commonly used parameter in clinical routine [6] and it is the reference parameter used for the definition of acute kidney injury by the KIDGO acute kidney injury guidelines [13]. Third, the range of ischemia duration and blood loss in this study was moderate, i.e., we cannot draw any conclusions on a potential interaction between ischemia duration and blood loss in extreme situations; however, the incidence of major events such as massive bleeding and very protracted ischemia durations can be expected to be low during clinical routine.

## 5. Conclusions

Preoperative renal function, ischemia duration, and blood loss, as markers for hypovolemia and hypotension, had a similar impact on postoperative function, albeit without

potentiating each other. While the surgical technique and ischemia minimization remain crucial to postoperative kidney function, increased awareness of conscious hemodynamic management appears to be warranted.

**Author Contributions:** Conception or design of the work: S.B. and A.H.; Data collection: R.M., E.M. and S.F.; Data analysis and interpretation: S.B.; Drafting of the article: S.B.; Critical revision of the article: R.M., E.M., A.M., S.F. and A.H.; Final approval of the version to be published: S.B., R.M., E.M., A.M., S.F. and A.H. All authors have read and agreed to the published version of the manuscript.

**Funding:** This research received no external funding.

**Institutional Review Board Statement:** Ethical review and approval were waived for this study according to According to Paragraphs 36 and 37 of the Hospital Law of the German region Rheinland-Pfalz ("Landeskrankenhausgesetz", § 36 and § 37), ethical approval and informed consent is not re-quired for retrospective analyses of routinely collected data as long as anonymity is preserved.

**Informed Consent Statement:** Not applicable.

**Data Availability Statement:** Source data are available upon reasonable request to the authors.

**Conflicts of Interest:** S.B. acts as proctor for the DaVinci Si-HD® System, Intuitive Surgical Inc., Sunnyvale, CA, USA; R.M., E.M., A.M., S.F. and A.H. do not report any conflicts of interest or competing interests.

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
