# Peer review of "Impact of Blood Loss on Renal Function and Interaction with Ischemia Duration after Nephron-Sparing Surgery"

_curroncol, doi:10.3390/curroncol29120767_

Round 1
Reviewer 1 Report
The current study aims to evaluate the impact of ischemia duration and blood loss on postoperative renal function in patients undergoing NSS. The authors should be congratulated for addressing a such an interesting topic. A major revision is needed. Please see below:
- Can you provide a table? For baseline characteristics. May you add some important information as eGFR?
- [An open technique was conducted in 321 (67.1%) patients] - What about the rest? Please clarify.
- [postoperative Hb] - When did you collect this data? At discharge, 1 post-operative day?
- Same question for postoperative creatinine. When did you collect it?
- Long-term renal function after PN is difficult to predict as it is influenced by several modifiable and nonmodifiable variables. Please include this recent reference [PMID: 35495283; PMCID: PMC9051959; ] in your manuscript where eGFR, sex, ischemia technique, and PPL are the best predictors of eGFR percentage loss at 1 yr after minimally invasive PN. This is worth to be mentioned in your discussion.
- Check typos.
Author Response
REVIEWER 1
The current study aims to evaluate the impact of ischemia duration and blood loss on postoperative renal function in patients undergoing NSS. The authors should be congratulated for addressing a such an interesting topic. A major revision is needed. Please see below:
We thank the reviewer for the kind appreciation of the importance of the topic. The thoughtful suggestions have been implanted and helped to substantially improve our manuscript.
- Can you provide a table? For baseline characteristics. May you add some important information as eGFR?
A baseline characteristics table was added as Table 1. Preoperative and postoperative estimated glomerular filtration rate are now reported as follows:
- page: “Preoperative median estimated glomerular filtration rate was 76.8 mL/min (IQR 59.2-91.5).”
- page: “Postoperative estimated glomerular filtration rate was 68.5 min/mL (IQR 49.8-86.6).”
- [An open technique was conducted in 321 (67.1%) patients] - What about the rest? Please clarify.
Information on the surgical technique was added as follows:
- page: An open technique was conducted in 321 (67.1%) patients. In 10 (2.1%) patients, the surgical technique was conventional laparoscopy, in 147 (30.8%) a robotic-assisted technique.”
- [postoperative Hb] - When did you collect this data? At discharge, 1 post-operative day? Same question for postoperative creatinine. When did you collect it?
We now specify in the text the time point of postoperative and creatinine measurement as follows:
- page: “Postoperative creatinine used was highest available value (peak) before discharge.”
- page: “Postoperative hemoglobin refers to the sample of the first postoperative day.”
- Long-term renal function after PN is difficult to predict as it is influenced by several modifiable and nonmodifiable variables. Please include this recent reference [PMID: 35495283; PMCID: PMC9051959; ] in your manuscript where eGFR, sex, ischemia technique, and PPL are the best predictors of eGFR percentage loss at 1 yr after minimally invasive PN. This is worth to be mentioned in your discussion.
The suggested reference was added in the introduction and in the discussion as follows:
- page : “The impact of postoperative renal functional impairment was confirmed in a recent study comparing multiple models for prediction of renal function impairment estimated by glomerular filtration rate decline at 1 year after partial nephrectomy: postoperative percentage glomerular filtration rate loss was a component among the 44 best predictive models, along with ischemia technique and preoperative glomerular filtration rate. “
- page: “Of note postoperative renal function impairment was recently demonstrated as major factor for decline of glomerular filtration rate at 1 year after partial nephrectomy.1
- Check typos.
Checked.
REVIEWER 1
The current study aims to evaluate the impact of ischemia duration and blood loss on postoperative renal function in patients undergoing NSS. The authors should be congratulated for addressing a such an interesting topic. A major revision is needed. Please see below:
We thank the reviewer for the kind appreciation of the importance of the topic. The thoughtful suggestions have been implanted and helped to substantially improve our manuscript.
- Can you provide a table? For baseline characteristics. May you add some important information as eGFR?
A baseline characteristics table was added as Table 1. Preoperative and postoperative estimated glomerular filtration rate are now reported as follows:
- page: “Preoperative median estimated glomerular filtration rate was 76.8 mL/min (IQR 59.2-91.5).”
- page: “Postoperative estimated glomerular filtration rate was 68.5 min/mL (IQR 49.8-86.6).”
- [An open technique was conducted in 321 (67.1%) patients] - What about the rest? Please clarify.
Information on the surgical technique was added as follows:
- page: An open technique was conducted in 321 (67.1%) patients. In 10 (2.1%) patients, the surgical technique was conventional laparoscopy, in 147 (30.8%) a robotic-assisted technique.”
- [postoperative Hb] - When did you collect this data? At discharge, 1 post-operative day? Same question for postoperative creatinine. When did you collect it?
We now specify in the text the time point of postoperative and creatinine measurement as follows:
- page: “Postoperative creatinine used was highest available value (peak) before discharge.”
- page: “Postoperative hemoglobin refers to the sample of the first postoperative day.”
- Long-term renal function after PN is difficult to predict as it is influenced by several modifiable and nonmodifiable variables. Please include this recent reference [PMID: 35495283; PMCID: PMC9051959; ] in your manuscript where eGFR, sex, ischemia technique, and PPL are the best predictors of eGFR percentage loss at 1 yr after minimally invasive PN. This is worth to be mentioned in your discussion.
The suggested reference was added in the introduction and in the discussion as follows:
- page : “The impact of postoperative renal functional impairment was confirmed in a recent study comparing multiple models for prediction of renal function impairment estimated by glomerular filtration rate decline at 1 year after partial nephrectomy: postoperative percentage glomerular filtration rate loss was a component among the 44 best predictive models, along with ischemia technique and preoperative glomerular filtration rate. “
- page: “Of note postoperative renal function impairment was recently demonstrated as major factor for decline of glomerular filtration rate at 1 year after partial nephrectomy.1
- Check typos.
Checked.
Reviewer 2 Report
In this paper, the authors tried to assess that ischemia duration and blood loss significantly affect postoperative renal function in patients undergoing NSS. The topic is actual and worthy of attention. However, a major revision is required.
· Please provide a baseline characteristics table
· When using a generalized linear model, please provide a graphic/table representation of the results.
· When presenting results. First baseline characteristics, perioperative data, and at the end histologic features. Please restructure the results section.
· You said, “DaVinci technique”. Please consider reformulating with “robotic approach”.
· 20 (4.2%) died. How? Cancer-related? Clarify.
· You mentioned Trifecta. Why? Did you assess it? If yes, better describe how and results; if not please eliminate it.
· To better describe renal function, please consider including also eGFR.
· The debate “clamp versus off-clamp” RAPN is still crucial. In general, it is widely accepted that limited ischemia time corresponds to preserved kidney function. Percutaneous tumor ablation (PTA) is recognized by current guidelines as a safe NSS option. Comparing RAPN vs PTA means a comparison between hilum mobilization/WIT vs no kidney damage/manipulation. Recent studies (RAPN vs PTA) reported worse Δ renal function at 1-yr for RAPN, while no difference at the Δ renal function at latest F/Up, both for single (PMID: 36216659; DOI: 10.1016/j.ejso.2022.09.022) and bilateral kidneys (PMID: 36367175; DOI: 10.1089/end.2022.0478). This means that we have kidney recovery after a while. This is worthy of interest and needs to be included in your discussion. I strongly suggest including those references.
· Make a conclusion paragraph.
Author Response
REVIEWER 2
In this paper, the authors tried to assess that ischemia duration and blood loss significantly affect postoperative renal function in patients undergoing NSS. The topic is actual and worthy of attention. However, a major revision is required.
We thank the reviewer for the kind appreciation of the importance of the topic. The thoughtful suggestions have been implanted and helped to substantially improve our manuscript.
Please provide a baseline characteristics table
A baseline characteristic table was added as Table 1.
When using a generalized linear model, please provide a graphic/table representation of the results.
The results are now reported in Table 2.
When presenting results. First baseline characteristics, perioperative data, and at the end histologic features. Please restructure the results section.
The results section was restricted as recommended. For the sake of readability, we did not copy/paste this section because submitted to major restructuring. We kindly refer to the tracked changes in the manuscript.
You said, “DaVinci technique”. Please consider reformulating with “robotic approach”.
Changed to “Robot-assisted technique”.
20 (4.2%) died. How? Cancer-related? Clarify.
Cause of death was specified as follows:
- page: “After a mean follow-up of 15 months, 29 (6.1%) patients developed distant metas-tasis and 20 (4.2%) died. Of the 20 deaths, 7 (1.5%) resulted from cancer, and 13 (2.7%) were not cancer related. Patients dying of cancer had in 71% (5/7) multilocular metastatic disease whereas metastasis was not present in any of the 13 patients dying from other causes.”
You mentioned Trifecta. Why? Did you assess it? If yes, better describe how and results; if not please eliminate it.
In line with Hung AJ J Urol. 2013. January;189(1):36–42 we referred to Trifecta as consissting in the three quality criteria of negative margins, no postoperative complications, and minimal renal functional loss. In the manuscript, data are provided for surgical margins, complication rate, and renal function. However, we agree with the Reviewer that the term appears only in the discussion and the data is presented as complication rate instead of lack of complication and proportion of positive margins instead of negative margins. Therefore, the corresponding section was reworded as follows:
- page: “We are aware of the following limitations. First, the data was generated at a single in-stitution, however relevant quality criteria like complication rate, surgical margins and renal function impairment Trifecta results as well as and oncological outcome was in the range of previously published data supporting the generalizability of our results”
To better describe renal function, please consider including also eGFR.
Preoperative and postoperative estimated glomerular filtration rate are now reported as follows:
- page: “Preoperative median estimated glomerular filtration rate was 76.8 mL/min (IQR 59.2-91.5).”
- page: “Postoperative estimated glomerular filtration rate was 68.5 min/mL (IQR 49.8-86.6).”
The debate “clamp versus off-clamp” RAPN is still crucial. In general, it is widely accepted that limited ischemia time corresponds to preserved kidney function. Percutaneous tumor ablation (PTA) is recognized by current guidelines as a safe NSS option. Comparing RAPN vs PTA means a comparison between hilum mobilization/WIT vs no kidney damage/manipulation. Recent studies (RAPN vs PTA) reported worse Δ renal function at 1-yr for RAPN, while no difference at the Δ renal function at latest F/Up, both for single (PMID: 36216659; DOI: 10.1016/j.ejso.2022.09.022) and bilateral kidneys (PMID: 36367175; DOI: 10.1089/end.2022.0478). This means that we have kidney recovery after a while. This is worthy of interest and needs to be included in your discussion. I strongly suggest including those references.
The suggested reference was added as follows:
- page: “However, as suggested in a study comparing robot-assisted partial nephrectomy and percutaneous tumor ablation with a follow-up of 3 years or more, impaired 1-year renal function may show some recovery over time.”
Make a conclusion paragraph.
Done.
Reviewer 3 Report
The paper addresses the issue of kidney damage as a result of ischaemia. Although the paper deals with renal oncological surgery, the authors rightly point out that the problem also occurs with other procedures involving blood loss, hypotensia and hypovolemia. Study group is sufficient, but for better clarity of the work it would be appropriate to include the data in the form of a table (number of patients, gender distribution, etc.).The authors used units of mg/dL of blood creatinine levels. Although these units are used in clinical practice, they suggest using µmol/l. if only in addition in brackets.
Please state what the authors mean by the term post operative- was the blood tested immediately after surgery or for example 6 hours afterwards?
[127] there is- (43,7)- should be (43,7%)
Author Response
REVIEWER 3
The paper addresses the issue of kidney damage as a result of ischaemia. Although the paper deals with renal oncological surgery, the authors rightly point out that the problem also occurs with other procedures involving blood loss, hypotensia and hypovolemia. Study group is sufficient, but for better clarity of the work it would be appropriate to include the data in the form of a table (number of patients, gender distribution, etc.).
Baseline characteristics are new summarized in Table 1.
The authors used units of mg/dL of blood creatinine levels. Although these units are used in clinical practice, they suggest using µmol/l. if only in addition in brackets.
Creatinine in µmol/L was added as suggested:
- Page: “Median preoperative creatinine was 0.94 mg/dL (IQR 0.82-1.16; median 83 µmol/L, IQR 72-102), median postoperative creatinine was 1.04 mg/dL (IQR 0.85-1.34; median 92 µmol/L, IQR 75-118).
Please state what the authors mean by the term post operative- was the blood tested immediately after surgery or for example 6 hours afterwards?
We now specify in the text the time point of postoperative and creatinine measurement as follows:
- - page: “Postoperative creatinine used was highest available value (peak) before discharge.”
- - page: “Postoperative hemoglobin refers to the sample of the first postoperative day.”
[127] there is- (43,7)- should be (43,7%)
Corrected.
Round 2
Reviewer 1 Report
I am satisfied with this new edition of your paper. I believe it is wontly of publication. Just a few concerns:
- You reported two times the reference n 8. Please correct it.
- Postoperative creatinine used was highest available value (peak) before discharge. Was done the same for eGFR? Please explain this. This is important because as you mentioned and reported the decline study of renal should be evaluated at 1 year after partial nephrectomy.
Author Response
I am satisfied with this new edition of your paper. I believe it is wontly of publication. Just a few concerns:- You reported two times the reference n 8. Please correct it.
This is now corrected (page 2)
- Postoperative creatinine used was highest available value (peak) before discharge. Was done the same for eGFR? Please explain this. This is important because as you mentioned and reported the decline study of renal should be evaluated at 1 year after partial nephrectomy.
The eGFR was estimated peak creatinine inhospital as suitable to the study question assessing the potential interaction between intraoperative blood loss/hypotension with ischemia duration in terms of acute kidney injury. However, we agree with the reviewer that GFR should also be followup closely in the long-term after partial nephrectomy. To exclude any misinterpretation with regard to timeline of eGFR the corresponding sentene now reads:
-page 4: “Postoperative inhospital estimated glomerular filtration rate was 68.5 min/mL (IQR 49.8-86.6).”
Reviewer 2 Report
Thank you for providing a new revised version of your paper. Its value is improved after revision. I appreciated the conclusion paragraph. I only have few questions and amendments:
- thank you for providing eGFR value in this new version. However, you did not clarify timing for postoperative eGFR. When did you collect it?
- You included in discussion the comparison among RAPN vs PTA and its evidence of no significant difference in Δ renal function at latest F/Up. This added great value to your manuscript because of the PTA spread in the last years. However, this is also true in the challenging scenario of RCC in solitary kidney. This population may is considered frail but RAPN such as PTA is possible with comparable functional outcomes. Please consider including this ref ((PMID: 36216659; DOI: 10.1016/j.ejso.2022.09.022) and consideration to better clarify functional outcomes vs one of the most valuable alternatives to surgery.
Author Response
Thank you for providing a new revised version of your paper. Its value is improved after revision. I appreciated the conclusion paragraph. I only have few questions and amendments:
-thank you for providing eGFR value in this new version. However, you did not clarify timing for postoperative eGFR. When did you collect it?
The eGFR was estimated peak creatinine inhospital as suitable to the study question assessing the potential interaction between intraoperative blood loss/hypotension with ischemia duration in terms of acute kidney injury. However, we agree with the reviewer that GFR should also be followup closely in the long-term after partial nephrectomy. To exclude any misinterpretation with regard to timeline of eGFR the corresponding sentene now reads:
-page 4: “Postoperative inhospital estimated glomerular filtration rate was 68.5 min/mL (IQR 49.8-86.6).”
- You included in discussion the comparison among RAPN vs PTA and its evidence of no significant difference in Δ renal function at latest F/Up. This added great value to your manuscript because of the PTA spread in the last years. However, this is also true in the challenging scenario of RCC in solitary kidney. This population may is considered frail but RAPN such as PTA is possible with comparable functional outcomes. Please consider including this ref ((PMID: 36216659; DOI: 10.1016/j.ejso.2022.09.022) and consideration to better clarify functional outcomes vs one of the most valuable alternatives to surgery.
We thank the Reviewer for this additional reference suggestion; however, we do not feel that any further extension of the comparison of RAPN vs PTA is warranted since it does not fit the topic of the study. Therefore, unless requested to do so by the Editor, we would prefer not to add further reference to PTA.
Reviewer 3 Report
Once corrected, the work can be published
Author Response
Once corrected, the work can be published.
Thank you for the appreciation of our revision efforts.
Round 3
Reviewer 1 Report
Thanks for providing a new revised version of the paper. I agree for pubblication.
Reviewer 2 Report
Thank you for clarification